# Whole Genome Sequencing of *Bacillus velezensis* AMR25, an Effective Antagonist Strain against Plant Pathogens

**DOI:** 10.3390/microorganisms12081533

**Published:** 2024-07-26

**Authors:** Alexey A. Ananev, Zlata V. Ogneva, Nikolay N. Nityagovsky, Andrey R. Suprun, Konstantin V. Kiselev, Olga A. Aleynova

**Affiliations:** Laboratory of Biotechnology, Federal Scientific Center of the East Asia Terrestrial Biodiversity, Far Eastern Branch of the Russian Academy of Sciences, 690022 Vladivostok, Russia; ananev.all@yandex.ru (A.A.A.); niknit1996@gmail.com (N.N.N.); suprun@biosoil.ru (A.R.S.); kiselev@biosoil.ru (K.V.K.); aleynova@biosoil.ru (O.A.A.)

**Keywords:** antimicrobial activity, genomic analysis, endophytes, *Botrytis cinerea*, *Fusarium oxisporum*, *Vitis amurensis*

## Abstract

The most serious problems for cultivated grapes are pathogenic microorganisms, which reduce the yield and quality of fruit. One of the most widespread disease of grapes is “gray mold”, caused by the fungus *Botrytis cinerea*. Some strains of *Bacillus*, such as *Bacillus halotolerans*, *Bacillus amyloliquefaciens*, and *Bacillus velezensis*, are known to be active against major post-harvest plant rots. In this study, we showed that the endophytic bacteria *B. velezensis* strain AMR25 isolated from the leaves of wild grapes *Vitis amurensis* Rupr. exhibited antimicrobial activity against grape pathogens, including *B. cinerea*. The genome of *B. velezensis* AMR25 has one circular chromosome with a length of 3,909,646 bp. with 3689 open reading frames. Genomic analysis identified ten gene clusters involved in the nonribosomal synthesis of polyketides (macrolactin, bacillene, and difficidin), lipopeptides (surfactin, fengycin, and bacillizin), and bacteriocins (difficidin). Also, the genome under study contains a number of genes involved in root colonization, biofilm formation, and biosynthesis of phytohormones. Thus, the endophytic bacteria *B. velezensis* strain AMR25 shows great promise in developing innovative biological products for enhancing plant resistance against various pathogens.

## 1. Introduction

Diseases caused by plant pathogens typically result in severe yield and quality losses in many economically important crops [1]. With the current trend to reduce the use of chemical pesticides due to their negative impacts on the environment and human safety, biocontrol is becoming increasingly attractive in agriculture. Biocontrol, which involves the use of biological control agents and their active compounds, is considered a sustainable approach to controlling plant diseases [2,3,4].

The endophytic microbiome is an important source of biocontrol agents. Plant endophytes are microorganisms that, under certain conditions, inhabit the tissues of living plants without having a negative impact on their function and development [5]. Endophytes are capable of producing compounds that are beneficial to the plant and producing substances that suppress the growth and development of pathogens [6]. Also, endophytic microorganisms promote growth and strengthen the plant’s immune system [7]. Biological preparations based on endophytic bacteria have shown promise for use in agriculture to control plant diseases [8]. The main assumption is that the relationship between endophytes and host plants is based on the interaction of rhizosphere microorganisms that stimulate plant growth [9]. Plant growth-promoting rhizobacteria (PGPRs) are bacteria that live in the rhizosphere and can improve the quality of plant growth either directly, by releasing plant growth-promoting substances, or indirectly, by preventing the harmful effects of the pathogen, improving the quality of plant growth [10]. In recent decades, several bacteria of the genera *Arthobacter*, *Azospirillum*, *Azotobacter*, *Bacillus*, *Burkholderia*, *Enterobacter*, *Klebsiella*, and *Pseudomonas* have been reported to increase the adaptability of plant pathogens [11].

*Bacillus* strains are considered important PGPRs, producing a wide range of bioactive secondary metabolites that can potentially inhibit the growth of plant pathogens and harmful rhizosphere microorganisms due to their ability to produce secondary metabolites [12]. These substances act against fungi such as *Botrytis* spp. and *Fusarium* spp., and pathogenic bacteria such as *Erwinia* spp. [13] Some strains of *B. velezensis* also produce secondary metabolites that exhibit antagonistic activity against various phytopathogens [14,15,16]. However, for a deeper understanding of the mechanisms of action of these substances and their practical application, additional research is required. Studies have shown that some strains of *B. velezensis* are important endophytes and have significant biological activity. They can suppress pathogens and stimulate plant growth [17,18,19]. There are several products based on *Bacillus* spp. for sale, such as Taegro^®^ (*Bacillus subtilis* var. *amyloliquefaciens* FZB24; Novozymes Biologicals, Inc., Salem, VA, USA) and Kodiak^®^ (*B. subtilis*; Bayer CropScience, St. Louis, MO, USA), which can be useful in agriculture [20].

An interesting approach to biocontrol is to isolate and characterize the endophytic microbiome of wild plants and then use individual microorganisms as biocontrol agents to prevent or treat plant diseases. *Vitis amurensis* Rupr. is the most northerly of the wild grapevines, growing in harsh climates. It has a high natural resistance to abiotic and biotic stresses, and its endophytic microbiome is a rich source of biocontrol agents [21]. The aim of this study was to determine the potential of the endophytic bacterium *Bacillus velezensis* strain AMR25, isolated from *V. amurensis* tissues, as a biological control agent against plant pathogens.

## 2. Materials and Methods

### 2.1. Bacterial Strains

*B. velezensis* AMR25 was isolated from the xylem of *V. amurensis* and stored in the collection (−80 °C) of the Biotechnology Laboratory of the Federal Scientific Center for Biodiversity FEB RAS, Vladivostok, Russian Federation. The complete genome has been sequenced and published in GenBank (acc. no CP140115.1). The genome sequences of the strains *B. amyloliquefaciens* N2B2 (GenBank acc. no GCA_030284565.1), *B. velezensis* FZB42 (GenBank acc. no GCA_000015785.2), *B. amyloliquefaciens* DSM7 (GenBank acc. no GCA_000196735.1), and *B. subtilis* 168 (GenBank acc. no GCA_000009045.1) were downloaded from the NCBI GenBank server (https://www.ncbi.nlm.nih.gov/, accessed on 7 May 2024).

### 2.2. Antagonism of Bacillus velezensis AMR25 on Petri Plates

Antibacterial activity was evaluated against bacterial plant pathogens such as *Erwinia billingiae* (GenBank acc. no MZ424741.1), *Pantoea agglomerans* (GenBank acc. no MZ424742.1), and *Xanthomonas campestris* (GenBank acc. no MZ424744.1) using the agar slab method [22]. The bacterial pathogens were identified through molecular techniques, which involved the amplification of bacterial 16S rRNA gene sequences using universal bacterial primers to produce 1500 bp 16S PCR products [23]. All plant pathogens utilized were isolated from the wild grapevine *V*. *amurensis* and added to the collection of microorganisms. The Laboratory of Biotechnology at the Federal Scientific Center of Biodiversity FEB RAS maintains a collection of several hundreds of diverse microorganisms, including bacteria and fungi, which have been isolated from various plant parts and the rhizosphere. The collection includes both endophytic and pathogenic microorganisms.

An overnight culture of *B*. *velezensis* AMR25 with a concentration of 3 × 10^8^ CFU/mL was applied to Petri dishes containing R2A medium (composition (g/L): proteose peptone, 0.5 casein hydrolyzed: 0.5 yeast extract: 0.5 dextrose: 0.5 starch: 0.5 sodium pyruvate: 0.3 di-potassium hydrogen phosphate: 0.3 magnesium sulfate: 0.024 agar: 15.0 pH: 7.2 ± 0.2) (PanReac, AppliChem, Darmstadt, Germany) and incubated at 26 °C for 12 h in the dark. Then, using a cork borer, agarose blocks with a diameter of 0.6 cm were cut from the dish with *B*. *velezensis* AMR25 and placed on Petri dishes with R2A medium, which had been inoculated with phytopathogenic bacterial strains. After incubation at 26 °C for 48 h, the areas of growth inhibition were measured using a caliper and the disc diameter was subtracted from the result [24]. To compare the antagonistic activity of *B. velezensis* AMR25 against different bacterial pathogen strains, the following criteria were used: growth inhibition diameter greater than 16 mm—very strong inhibition, 11–15.9 mm—strong inhibition, 6–10.9 mm—moderate inhibition, 1–5.9 mm- weak inhibition, 0 mm—no inhibition [22].

The fungicidal activity test was carried out using the dual culture method on R2A medium against the pathogenic fungi *B. cinerea* (GenBank acc. no KP151610.1), *Fusarium avenaceum* (GenBank acc. no OR591465.1), and the opportunistic plant pathogen *Alternaria tenuissima* (GenBank acc. no MZ427922). An isolate of the pathogenic fungus was obtained from an infected leaf of *V. amurensis* and the species identification was carried out by a molecular genetics method using the *ITS1* region, and amplified sequences using universal primers [25]. *B. velezensis* was placed on a Petri dish at a distance of 2 cm from the pathogen. The inhibition area was measured on the 5th day of cultivation at 26 °C. The inhibition index was calculated using the following formula:(R1 − R2)/R1 × 100
where R1 is the radius of colony growth in the direction opposite to the candidate microorganism, and R2 is the radius of colony growth in the direction away from the candidate microorganism [26].

The experiments were performed at least three times and the results are represented as mean ± standard error.

### 2.3. Antibiotic Resistance Analysis of Bacillus velezensis AMR25

The *B. velezensis* AMR25 resistance testing was performed using the Kirby–Bauer disc diffusion method in accordance with the Clinical and Laboratory Standards Institute (CLSI) standards for antimicrobial resistance testing [27]. Thirteen antibiotics were tested: amoxicillin (10 μg/mL), penicillin (10 μg/mL), azithromycin (10 μg/mL), gentamicin (10 μg/mL), ceftazidime (30 μg/mL), vancomycin (30 μg/mL), erythromycin (15 μg/mL), oleandomycin (15 μg/mL), levofloxacin (5 μg/mL), ofloxacin (5 μg/mL), cefoperazone (75 μg/mL), oxacillin (1 μg/mL), and tetracycline (30 μg/mL). The discs were manufactured by the Azimut Company (Saint-Petersburg, Russia). The inoculated disks were incubated for 16 h at 26 °C. Inhibition areas were measured in millimeters using a caliper.

### 2.4. Genome Sequencing, Assembly and Annotation of Bacillus velezensis AMR25

A single colony of AMR25 was placed on a Petri dish containing R2A agar medium and grown overnight at 28 °C. The overnight bacterial suspension was centrifuged at +4 °C 6000× *g* for 6 min (5415R, Eppendorf, Leipzig, Germany), then the supernatant was removed from the test tube and the bacterial precipitate was used to isolate DNA. Total DNA was isolated by the hexadecyltrimethylammonium bromide (CTAB) method with modifications [21,28] followed by reprecipitation in 1 M sodium acetate. The concentration of the resulting DNA was determined using the NanoPhotometer P300 (IMPLEN, Munich, Germany). Whole genome sequencing was performed by “GENOANALYTICA” company (Moscow, Russian) using the MinION (Oxford Nanopore Technologies, Oxford Science Park, Oxford, UK). To create the circular structure of the chromosome from individual fragments (contigs), the Unicycler program version 0.5.0 was used [29]. The BUSCO program was used to assess the quality of the assembly [30]. The NCBI Prokaryotic Genomes Automatic Annotation Pipeline (PGAAP) was used to automatically annotate the *B. velezensis* AMR25 bacterial genome. The functional description of the genome was performed by comparing the nucleotide sequence obtained with known sequences available in databases such as “Clusters of Orthologous Groups of Proteins” (COG) and “Gene Ontology” (GO) resources. To annotate the genes, the researchers used the eggNOG (version 4.5) resource in combination with the eggNOG-mapper (version 2.1.12) [31].

### 2.5. Phylogenetic Analysis

Phylogenetic analysis was performed using the DNA gyrase subunit A (*gyrA*) gene sequence. NCBI BLAST software (accessed on 3 May 2024) [32] was used to determine the *Bacillus* species closest to endophytic strain AMR25. The following sequences were selected for phylogenetic analysis, which corresponded to 135–986 bp of the *gyrA* gene: *B. velezensis* strain AMR25 (CP140115.1:6730-7580), *B. subtilis* subsp. *subtilis* str. 168 (NC_000964.3:7129-7979), *B. velezensis* strain FZB42 (NC_009725.2:7138-7988), *B. siamensis* KCTC 13613 (NZ_AJVF01000039.1:399-1249), *B. velezensis* strain ZY1 (NZ_CP117945.1:6730-7580), *B. amyloliquefaciens* strain DSM7 (NC_014551.1:7145-7995), *B. amyloliquefaciens* strain LM2303 (NZ_CP018152.1:1324081-1323231), *B. velezensis* strain FJAT-45028 (NZ_CP047157.1:6730-7580), and *B. amyloliquefaciens* strain N2B2 (NZ CP127243.1:3914698-3915548). The *Bacillus* neighbor-joining tree was constructed using partial *gyrA* nucleotide sequences, with bootstrap support levels based on neighbor-joining analyses of 1000 resampled data sets.

### 2.6. Analysis of Average Nucleotide Identity

Average Nucleotide Identity (ANI) is a simple algorithm that simulates DNA-DNA hybridization (dDDH) [33]. OrthoANI takes into account the concept of orthology, where both genome sequences were fragmented and only pairs of orthologous fragments were used to calculate nucleotide identity. The genomes of the sequenced *Bacillus* strains were compared with the *B. velezensis* AMR25 genome using the OrthoANI average nucleotide identity analysis tool [34].

### 2.7. Comparative Genomics

The complete genome of *B. velezensis* AMR25 has been sequenced and published in GenBank (CP140115.1). The genome sequences of the strains *B. amyloliquefaciens* N2B2 (GenBank GCA_030284565.1), *B. velezensis* FZB42 (GenBank GCA_000015785.2), *B. amyloliquefaciens* DSM7 (GenBank GCA_000196735.1), and *B. subtilis* 168 (GenBank GCA_000009045.1) were downloaded from the NCBI GenBank server.

The genes responsible for the production of transport and ribosomal RNAs were identified using the tRNAscan-SE (version 2.0) and RNAmmer (version 1.2) programs, respectively. The online platform ISFinder (http://www-is.biotoul.fr/, accessed on 5 April 2024) was used to determine the presence of the insertion sequence (Insertion Sequence, IS). The PHASTER online service was used to find phages in the genome [35]. Circular maps were generated using BLAST Ring ImageGenerator (BRIG) (version 0.95) [36]. The Resistance Gene Identifier (RGI) (version 6.0.3) was used to search for resistance genes [37].

### 2.8. Secondary Metabolites Analysis

AntiSMASH 7.0.0 software [38] was used to identify the gene cluster associated with secondary metabolites *B. velezensis* AMR25 genome. AntiSMASH 7.0.0 with default settings was used to detect the structure of secondary metabolism against MIBiG. OrthoVenn3 software [39] was used to create a Venn diagram and find the unique gene clusters of *B. velezensis* AMR25.

## 3. Results

### 3.1. Antipathogenic Activity of Bacillus velezensis AMR25

Using the agar slab method to study the antibacterial activity of *B*. *velezensis* AMR25 on *E*. *billingiae*, *P*. *agglomerans*, and *X*. *campestris* (Figure 1) showed that it exhibited weak antibacterial activity. The inhibition area was 4.33 ± 0.33, 3.66 ± 0.33, and 1.66 ± 0.33 mm, respectively (Table 1).

To analyze the fungicidal activity, the dual culture method was used against pathogens such as *A*. *tenuissima*, *B*. *cinerea*, and *F*. *avenacea* (Figure 2). The AMR25 strain showed the strongest inhibitory activity against *B. cinerea* at 71.72 ± 1.48% (Table 2). Against *A*. *tenuissima* and *F*. *avenacea*, the fungicidal activity was lower at 65.71 ± 1.48% and 62.86 ± 1.48%, respectively (Table 2, Figure 2).

### 3.2. Genome Features

The *B. velezensis* AMR25 genome was sequenced using the Oxford Nanopore technology. After quality control, 3872 MB of data were collected and analyzed, resulting in one scaffold. The main features of the AMR25 genome are shown in Figure 3. It contains a circular chromosome of 3,909,646 base pairs, with an average guanine–cytosine composition of 46.61% in the DNA (guanine–cytosine ratio). In addition, 3716 CDS genes with an average length of 923 base pairs were found, with protein-coding regions accounting for 88.50% of the total genome. There were 86 tRNA genes and nine rRNA operons were identified.

### 3.3. Phylogenetic Analysis of Bacillus velezensis AMR25

The housekeeping gene *gyrA*, encoding the subunit A of DNA gyrase (*gyrA*), is essential for DNA replication and is found in all bacteria [40]. *gyrA* identity analysis provides higher phylogenetic resolution than 16S rRNA gene analysis for *Bacillus* species [41,42]. Specifically, partial *gyrA* gene sequence analysis was used for phylogenetic and species identification analysis of seven *Bacillus* strains including *B. amyloliquefaciens*, etc. [28]; thus, a phylogenetic analysis based on *gyrA* gene sequences was performed and AMR25 was identified as a member of the species *B. velezensis* (currently, *B. methylotrophicus* and *B. amyloliquefaciens* subsp. *plantarum* are recognized as later heterotypic synonyms for *B. velezensis*) [43]. The closest strain whose genome is represented in GenBank was *B. amyloliquefaciens* N2B2 (Figure 4). Multiple alignments of the partial *gyrA* sequences from representatives of the *Bacillus* genus can be found in Supplementary File Appendix A. These data were also confirmed by the OrthoANIu and dDDH estimates (GLM-based), which were 99.38% and 95.10%, respectively. The proposed and generally accepted species boundaries for OrthoANIu and dDDH values are less than 95~96% and 70%, respectively.

### 3.4. Comparison of Genetic Characteristics of Differentd Bacillus Strains

The strains *B. amyloliquefaciens* N2B2 (the most closed endophytic strain), *B. velezensis* FZB42 (a well-known endophyte), *B. velezensis* DSM7 (a soil bacterium without endophytic properties), and *B. subtilis* subsp. *subtilis* 168 (a domestic strain) were selected for further analysis. General characteristics of these strains are presented in Table 3.

The OrthoVenn analysis revealed that *B*. *amyloliquefaciens* DSM7 contained 3605 gene clusters, *B*. *subtilis* 168 contained 3364 clusters, *B*. *velezensis* AMR25 contained 3614, *B*. *velezensis* FZB42 contained 3561 clusters, and *B*. *amyloliquefaciens* N2B2 contained 3668 genetic clusters (Figure 5). The OrthoVenn diagram demonstrated that all five *Bacillus* species shared 3027 gene clusters. A total of 3197 gene clusters were shared between AMR25 and *B*. *subtilis* 168, but only 4 of them were found in these two strains alone. AMR25 shared 3452 gene clusters with *B*. *velezensis* FZB42, 32 of which were found only in these two strains. A total of 3527 genetic clusters were found between AMR 25 and *B. amyloliquefaciens* N2B2, 32 of which were unique to these two strains. AMR25 shared a total of 3396 common genetic clusters with *B*. *amyloliquefaciens* DSM7, 9 of which were unique to these two strains. *B*. *velezensis* AMR25 also contained a single unique cluster represented by three genes (Figure 5).

Using the antiSMASH software, 12 families of genes involved in the synthesis of secondary metabolites were identified. Only eight of these families can be specifically correlated with genes responsible for the synthesis of specific compounds (Table 4 and Figure 6). These eight compounds can be divided into three main categories: polypeptides (e.g., andalusicin, bacillibactin, and bacilysin), lipopeptides (such as surfactin and fengycin), and polyketides (including macrolactin, bacillaene, and difficidin).

Of the eight secondary metabolites detected in strain AMR25, each has antibacterial activity. However, only three of them—surfactin, bacillaene and fengycin—also have fungicidal activity (Table 4).

### 3.5. Gene Clusters of Antipathogenic Activity Identified in the Genome of Bacillus velezensis AMR25

Comparative analysis of the gene clusters using antiSMASH 7.0.0 revealed minor differences between the endophytic *B. velezensis* strains AMR25, N2B2, and FZB42; a common feature of these strains was the presence of macrolactin and difficidin; the *B. velezensis* AMR25 strain was also characterized by the presence of an additional cluster of genes for the synthesis of the antibiotic andalusicin (Figure 7).

Of the eight secondary metabolites synthesized by strain *B. velezensis* AMR25 and discovered using antiSMASH 7.0.0, all have antibacterial activity according to the literature, but only two have fungicidal activity—surfactin and fengicin. It is noteworthy that in previous analysis, a cluster unique to *B. velezensis* AMR25, consisting of three genes, was discovered, which was not detected in the antiSMASH analysis. The detected genes correspond to mycosubtilin synthase subunits A, B, and C. Mycosubtilin is a hybrid natural product of trans-AT fatty acid synthase (FAS)/NRP.

The next step of this work was to determine the presence of all the genes necessary for the biosynthesis of fengycin, surfactin, and mycosubtilin in the cluster. Using the KEGG pathway database, it was shown that the AMR25 genome contains all the necessary genes (Figure 8a–c in red). It is noteworthy these two compounds have very similar structures and have the same mechanism of action. These compounds bind to sterols and form pores in the cell membranes of the pathogen, changing the permeability of the cell membrane and leading to cell death [53]. In addition, the presence of genes required for the biosynthesis of six other metabolites from *Bacillus velezensis* AMR25—bacillisin, bacillaene, bacillibactin, dificidin, macrolactin, and andalucin (see Figure 8d–i and Appendix A)—was analyzed. Using databases from the Kyoto Encyclopedia of Genes and Genomes (KEGG), genes encoding for compounds such as bacillin and bacillobactin are fully present in the *B. velezensis* AMR25 genome. For compounds such as bacillus and deficient, one gene is missing from each respective gene cluster. The products of these genes have not yet been registered in KEGG due to the unknown functions of these hypothetical proteins. Nevertheless, they are present in AMR25 gene clusters, although their specific roles remain to be determined. With regard to the andalucin molecule, there is currently very little information available regarding which enzymes are involved in its biosynthesis. It is understood that this involves a cluster of genes that contain the enzyme LanKC, which is a lanthionine synthetase of class III and various auxiliary proteins, such as the permease carrier ABC-2 and SAM-dependent methyltransferase class I, among others. It has been demonstrated that the gene that encodes the LanKC enzyme is present in the genome of *B. velezensis* AMP25 (see Appendix A).

### 3.6. Antibiotic Resistance Testing Endophytic Bacteria Bacillus velezensis AMR25

Analysis of antibiotic resistance showed that the *B. velezensis* AMR25 strain had multiple antibiotic resistance (Table 5); in particular, the strain was resistant to all studied beta-lactam antibiotics (penicillins and cephems), as well as to macrolides, aminoglycosides, and tetracyclines. With regard to the fluoroquinolone group, the strain showed resistance to ofloxacin and sensitivity to levofloxacin, in addition to intermediate values for vancomycin (Table 5).

A search of the CARD database for antibiotic resistance genes in the genome of *B. velezensis* AMR25 revealed the presence of 10 genes responsible for resistance to antimicrobial compounds (Table 6). Of these, eight were found to be responsible for resistance to various antibiotics, while the remaining two were identified as genes responsible for resistance to disinfectants and resistance to antiseptics. Two genes responsible for resistance to glycopeptide antibiotics, *vanT* and *vanY*, were identified among the genes responsible for antibiotic resistance. This finding aligns with the phenotype data presented in Table 5. Two additional genes, *BcI* and *ponA*, which are responsible for resistance to beta-lactams, and *msrA* and *msrB*, which are responsible for resistance to macrolide antibiotics, were also identified. These findings are consistent with the observed phenotype. The data on tetracycline resistance also correlate with the phenotype (Table 5). No genes responsible for gentamicin resistance were identified.

### 3.7. Genetic Elements of Bacillus velezensis AMR25 Responsible for Plant-Bacterial Interaction

#### 3.7.1. IAA Products

Endophytic bacteria, including certain strains of the genus *Bacillus*, are able to produce indole-3-acetic acid (IAA), which is necessary for plant development and plays an important role in plant–microbe interactions [54]. Keswani and colleagues [55] reported in 2020 that *Bacillus* bacteria have four different pathways for IAA synthesis, namely the indole-3-acetamide pathway (IAM), the indole-3-pyruvic acid (IPyA) pathway, the indol-3-acetonitrile pathway (IAN), and the IAA acetyltransferase (IAAT) pathway. The genome sequence of *B. velezensis* AMR25 contains genes corresponding to three pathways for the IAA biosynthesis, namely IPyA (*patB*, *yclC*, and *dhaS*), IAN (*yhcX*), and IAAT (*ysnE*) (Appendix A).

#### 3.7.2. Cytokinin Products

Another group of phytohormones involved in plant growth and development is the cytokinins [56]. The genome of *B. velezensis* AMR25 contains genes involved in cytokinin production, such as *miaA* encoding tRNA dimemthallyl transferase; *miaB* encoding 2-methylthio N6-dimemthallyladenosne synthase; *yvdD* encoding cytokinin monophosphate phosphoribo-hydrolase; and *dapF* encoding diaminopemlate epimerase (Appendix A). The discovery of these genes suggests that *B. velezensis* AMR25 is capable of producing cytokinins, which is consistent with previous reports of the ability of the genus *Bacillus* to synthesize cytokinins [57,58].

#### 3.7.3. Nutrient Acquisition

Phosphorus, nitrogen, and sulfur are essential nutrients for plant growth and development. The genome of *B. velezensis* AMR25 contains 15 genes responsible for nitrogen metabolism, including nitrate reductase (*narJ*, *narG*, *narH*, and *narI*) and nitrate/nitrite transport (*nsrR* and *narK*) (Appendix A). In addition, genes responsible for urease degradation (*ureC*, *ureB*, and *ureA*) were found (Appendix A). *Cys* genes involved in sulfate reduction and transport, and *ssu* genes involved in sulfonate transport and degradation, were found (Appendix A). The genes responsible for phosphate transport and assimilation are also present in the genome of AMR25 strain and are represented by 12 genes, including phosphate phosphatases (*phoD*, *ycsE*, and *rsbU* genes), phosphorus transporter genes (*pstABC*), and phytase (*phy*). The discovered genes suggest that *B. velezensis* AMR25 may contribute to increased nutrient production, which is consistent with previous reports [59,60].

#### 3.7.4. Motility, Chemotaxis and Adhesion of Plants

The genome of *B*. *velezensis* AMR25 contains a chemotaxis gene cluster responsible for recognizing the plant genes *cheABDRWY* and methyl acceptor chemotaxis proteins such as *mcpABC*, *hemA*, and *yfmS* (Appendix A). Also, the genome contained clusters of genes responsible for motor functions, such as *fliDEFGHIJKLMPQRSWYZ*, *flhABF*, and *flbD* (Appendix A) [59]. The attachment of bacteria to the surface of the plant is due to the formation of biofilm. A cluster of *epsDFGHIJKLMNO* genes responsible for increasing cell aggregation and the production of capsule polysaccharides was found in the AMR25 genome (Appendix A). Biofilm formation regulation genes (*ymcA*, *ylbF*, *sinR*, *rpoN, cysE,* and *csrA*) (Appendix A), as well as genes responsible for quorum sensing (*hfq*, *trpE, rfb*, and *LuxS*) (Appendix A) were also found [56].

## 4. Discussion

Biocontrol is a promising method to control various plant diseases caused by pathogens of both bacterial and fungal nature [2]. The use of bacteria of the genus *Bacillus* to protect plants from diseases is a promising direction in agriculture [20]. Endophytic bacteria, such as *Bacillus* sp., can protect plants from bacterial and fungal pathogens, stimulate plant growth, and improve nutrient availability [4,6]. *Bacillus* spp. possess a wide range of antimicrobial compounds and are capable of inducing plant responses to pathogens [7]. They are also resistant to various unfavorable environmental conditions and form spores, which facilitates the production of preparations for the treatment of plants [2,20].

Therefore, research into biocontrol using *Bacillus* sp. is relevant and has great potential for improving plant protection methods and increasing crop yields.

*Bacillus* is the prevalent type of bacterial endophyte capable of inhibiting fungal growth, which is consistent with the large number of *Bacillus* strains that have been described in the literature as fungal antagonists [61]. It is interesting to note that *B. methylotrophicus* (now *B. methylotrophicus* and *B. amyloliquefaciens* subspecies *plantarum* are recognized as heterotypic synonyms of *B. velezensis*) [43] is among the prominent types of endophytes present in Chinese ginseng *Panax notoginseng* [62]. Strains of *B. methylotrophicus* show a wide range of antagonistic activity against fungal pathogens including *Fusarium oxysporum*, *Bipolaris sorokiniana*, and *Meloidogyne hapla* [63,64]. Additionally, *B. methylotrophicus* strain BC79, which was isolated from forest soil, can colonize rice plants and displays 89.87% biocontrol efficacy against the rice blast pathogen *Magnaporthe oryzae* [64].

This study presents an analysis of the genome of the PGRP bacterium *B. velezensis* AMR25, an endophyte of the wild grapevine *V. amurensis*, with high antibacterial and fungicidal activity. Genes responsible for the biosynthesis of the antibiotic andalucinin were discovered, as well as genes for the biosynthesis of three fungicides—surfactin, fengycin, and mycosubtillin. It is known that surfactin is active against leaf spot caused by *Cochliobolus carbonum*, bacterial mottle caused by *Pseudomonas syringae* pv. tomato, potato brown rot caused by *Ralstonia solanacearum*, and the causal agent of corn ear *Fusarium verticillioides* [45]. Fengycin has antibacterial activity against potato brown rot caused by *Ralstonia solanacearum* and bacterial black spot caused by *Xanthomonas euvesicatoria* [50,51]. It is also effective against various fungal pathogens such as *Fusarium oxysporum* and *Fusarium graminearum*, and against gray mold caused by *B. cinerea* [49]. Bacillaene has been shown to be effective against bacterial fireblight in fruit crops caused by *Erwinia amylovora* [48]. Macrolactin was found to be an effective antibacterial against bacteria such as *Staphylococcus aureus* and *Burkholderia cepacia* [47]. Both difficidin and bacillusin show antibacterial activity against the pathogens that cause bacterial fireblight in fruit crops, which are caused by the bacteria *Xanthomonas oryzae* pv. oryzae and *Erwinia amylovora* [46]. The production of siderophores, which were found in the *B. velezensis* AMR25 genome, and is also associated with antifungal activity [65] as mentioned earlier, was found for the strain *B. velezensis* AMR25 under investigation. This may explain the high antagonistic activity of *Bacillus* AMR25 against *B. cinerea*, where the antagonism index was 71.72 ± 1.48%. The percentage inhibition found in this study is comparable to other reduction effects described for *Bacillus subtilis* and *Bacillus amyloliquefaciens* [45,49,66].

Antibiotic-producing bacteria have many complex self-defense mechanisms against antibiotics, including their own antibiotics. Very often they have several mechanisms in place at the same time, providing complete protection against the biologically active molecules they produce. In this study, the sensitivity of the AMR25 strain to commonly used antibiotics was tested. In this study, the *B. velezensis* strain AMR25 was found to be resistant to many antibiotics, and resistance genes were also found for most antibiotics. However, the AMR25 strain was resistant to gentamicin, but no resistance genes were found in the genome. It is known that target modification resistance to aminoglycosides uses 16S rRNA methyltransferases that methylate at residue A1408 or [67]. It has also been shown that the chromosomally encoded enzyme AAC(2′)-Ia from *Providencia stuartii* bacteria has the function of acetylating and recycling peptidoglycans, although it can also acetylate aminoglycosides. Thus, there may be “random” substrates for these enzymes due to their similarity to cellular substrates containing amino sugars [68]. These studies further illustrate the plasticity of antibiotic-modifying enzymes [69,70]. In addition, there are internal mechanisms of antibiotic resistance that are usually encoded by chromosomes and include non-specific efflux pumps, enzymes that inactivate antibiotics, or mechanisms that act as permeability barriers [71,72]. Also, the transfer of genes that make bacteria resistant to antibiotics happens through mechanisms of genetic exchange, collectively known as horizontal gene transfer (HGT) mechanisms [73,74]. Plasmid-mediated conjugation is known to be a much more common gene transfer mechanism than transformation or transduction for the spread of resistance genes in nature. Plasmids are capable of autonomous replication and carry a number of resistance genes as part of transposons, conferring resistance to several classes of antibiotics and metal ions simultaneously [75]. Therefore, the presence of AMR25 gentamicin resistance may be due to the presence of a plasmid containing aminoglycoside antibiotic resistance genes. Unfortunately, the genomic analysis of plasmids was not performed in this study, so this hypothesis requires further investigation.

Furthermore, analysis of the *B. velezensis* AMR25 genome revealed genes involved in plant–microbe interactions, including phytohormone biosynthesis, nitrogen and sulfur metabolism, phosphate solubilization, adhesion, and biofilm formation. A total of 32 proteins involved in chemotaxis and flagellar formation, and 21 genes involved in adhesion and biofilm formation, were discovered. These data suggest a mutually beneficial relationship between the bacterium *B. velezensis* AMR25 and the grapevine plant. Thus, *B. velezensis* AMR25 can be used to control fungal pathogens including *Fusarium* and *Botrytis* sp. for grapevines. Further studies are planned to investigate the fungicidal properties of *B. velezensis* AMR25 on grapevines, when plants are infected with *Botrytis* and *Fusarium*.

## 5. Conclusions

Whole genome sequencing, assembly, and annotation of the *B. velezensis* AMR25 genome was performed. The genome size was 3.9 Mb. According to the results of the phylogenetic analysis, strain *B. velezensis* AMR25 was assigned to the species *B. velezensis.* The endophytic bacterial strain *B. velezensis* AMR25 harbored nine distinct genetic clusters that were identified as the key players in the synthesis of antimicrobial compounds. Among these clusters, three were particularly noteworthy for their fungicidal properties: surfactin, fengycin, and mycosubtillin. Thus, the *B. velezensis* AMR25 endophytic bacterial strain isolated from *V. amurensis* is promising for the development of biological products to protect plants from fungal pathogens, such as *B. cinerea* and *F. oxysporum*.

The mechanism by which *B. velezensis* AMR25 affects the pathogenic fungus remains unknown, so further studies will involve mass spectrometry and transcriptomic analysis. Additionally, an in planta experiment is planned to determine the mechanisms of biological control of fungal infection in a diseased plant, as well as to compare the effectiveness of existing products based on other *Bacillus* strains with the *B. velezensis* AMP25 strain.

## Figures and Tables

**Figure 1 microorganisms-12-01533-f001:**
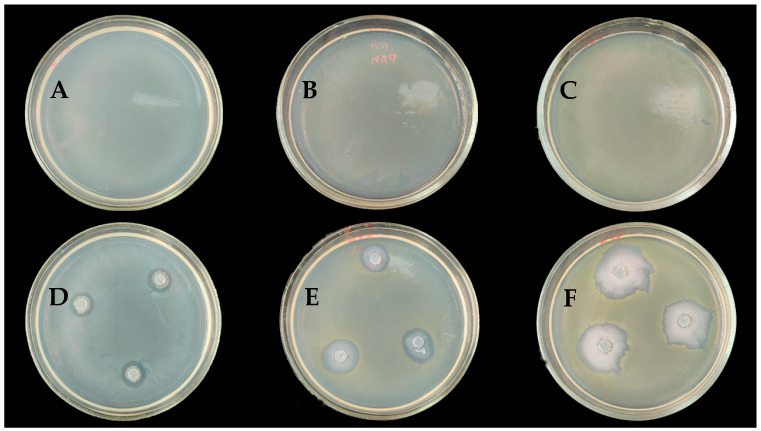
Antagonism of *Bacillus velezensis* AMR25 against *Erwinia billingiae* (**A**,**D**), *Pantoae agglomerans* (**B**,**E**), and *Xantomonas campestris* (**C**,**F**) using the dual culture plate assay. Inhibition of bacterial growth due to the action of bacteria. Petri dishes (**A**–**C**) are control, where *E. billingiae*, *P. agglomerans*, and *X. campestris* were placed, respectively, without the influence of the AMP25 strain. The diameter of the Petri dish 10 cm.

**Figure 2 microorganisms-12-01533-f002:**
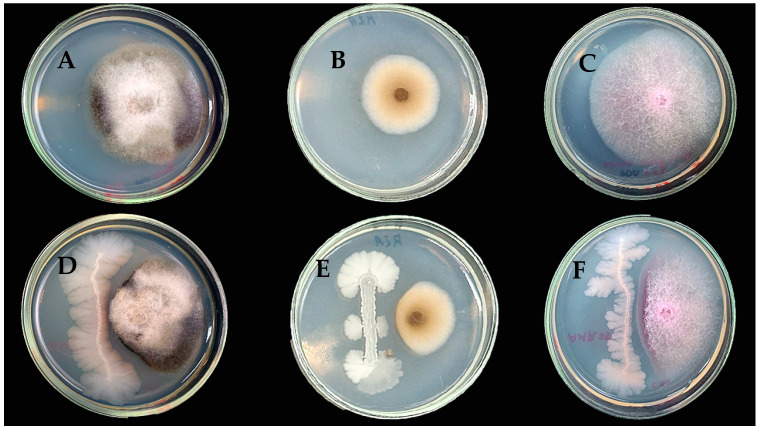
Antagonism of *Bacillus velezensis* AMR25 against *Alternaria tenuissima* (**A**,**D**), *Botrytis cynerea* (**B**,**E**), and *Fusarium avenacea* (**C**,**F**) using the dual culture plate assay. Reduction in fungi growth due to bacterium action. The diameter of the Petri plate is 10 cm.

**Figure 3 microorganisms-12-01533-f003:**
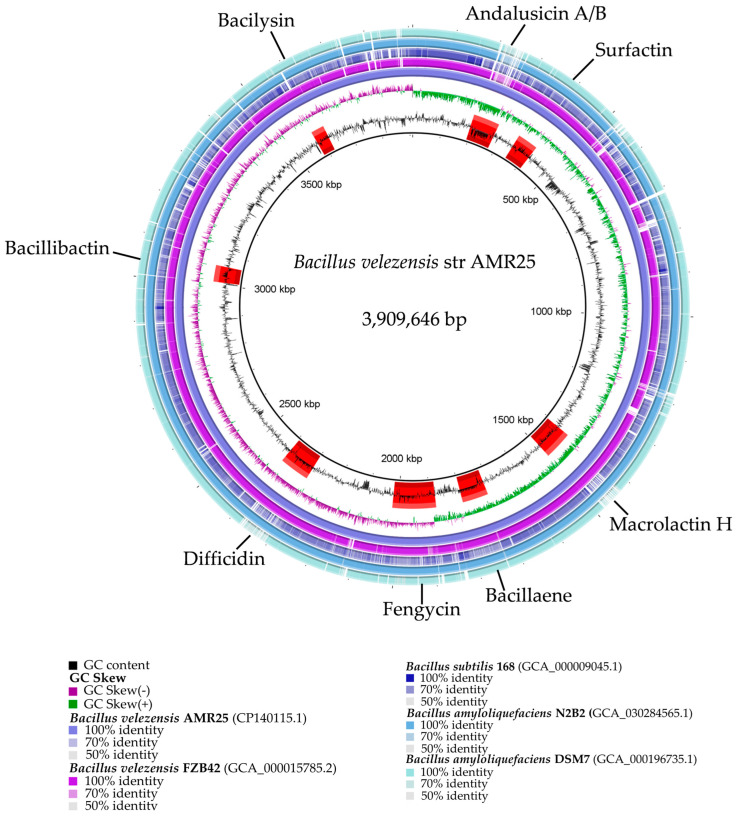
Genomic comparison of *Bacillus velezensis* AMR25 with closely related representative strains of *Bacillus*. These circular maps were created using the BLAST Ring Image Generator (BRIG) [36]. The concentric rings represent the sequences of each representative *Bacillus* strain in comparison with the reference strain, *B. velezensis* AMR25. The 1st ring (inner) shows the GC content of *B. velezensis* AMR25, and secondary metabolites with biologically controlled activity are marked on this ring by red arcs (andalusicin, surfactin, fengycin, bacillibactin, bacilysin, macrolactin, bacillaene, difficidin). The 2nd ring (middle) shows a general view of *B. velezensis* AMR25, while the 3–7th rings compare the genome sequence of AMR25 to genomes from *B. subtilis* 168, *B. amyloliquefaciens* DSM7, *B. velezensis* FZB42, and *B. velezensis* N2B2. Percentages show the degree of similarity between the reference genome and the other *Bacillus* genomes.

**Figure 4 microorganisms-12-01533-f004:**
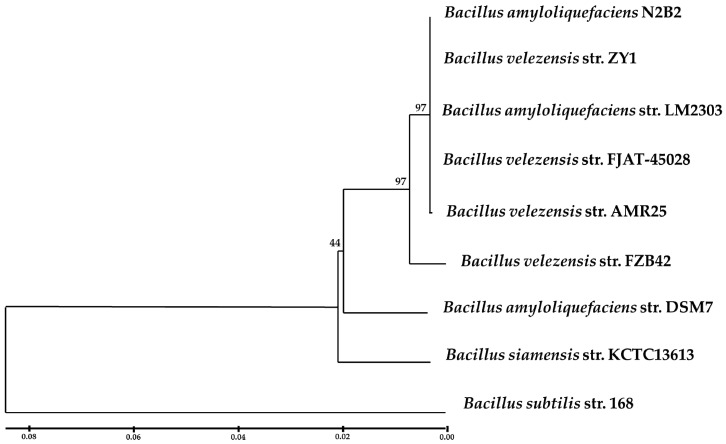
Neighbor-joining tree of *Bacillus velezensis* AMR25 based on partial *gyrA* sequences (nucleotides 135 to 986 of *gyrA*). The following sequences were selected for phylogenetic analysis: *Bacillus velezensis* strain AMR25 (CP140115.1:6730-7580), *Bacillus subtilis* subsp. *subtilis* str. 168 (NC_000964.3:7129-7979), *Bacillus velezensis* strain FZB42 (NC_009725.2:7138-7988), *Bacillus siamensis* KCTC 13613 (NZ_AJVF01000039.1:399-1249), *Bacillus velezensis* strain ZY1 (NZ_CP117945.1:6730-7580), *Bacillus amyloliquefaciens* strain DSM7 (NC_014551.1:7145-7995), *Bacillus amyloliquefaciens* strain LM2303 (NZ_CP018152.1:1324081-1323231), *Bacillus velezensis* strain FJAT-45028 (NZ_CP047157.1:6730-7580), and *Bacillus amyloliquefaciens* strain N2B2 (NZ CP127243.1:3914698-3915548). Percentages at nodes indicate bootstrap support levels based on neighbor-joining analysis of 1000 resampled data sets. The scale bar indicates 0.02 nucleotide substitutions per nucleotide position. Further information regarding the multiple alignment and the sequences utilized can be found in Appendix A, respectively.

**Figure 5 microorganisms-12-01533-f005:**
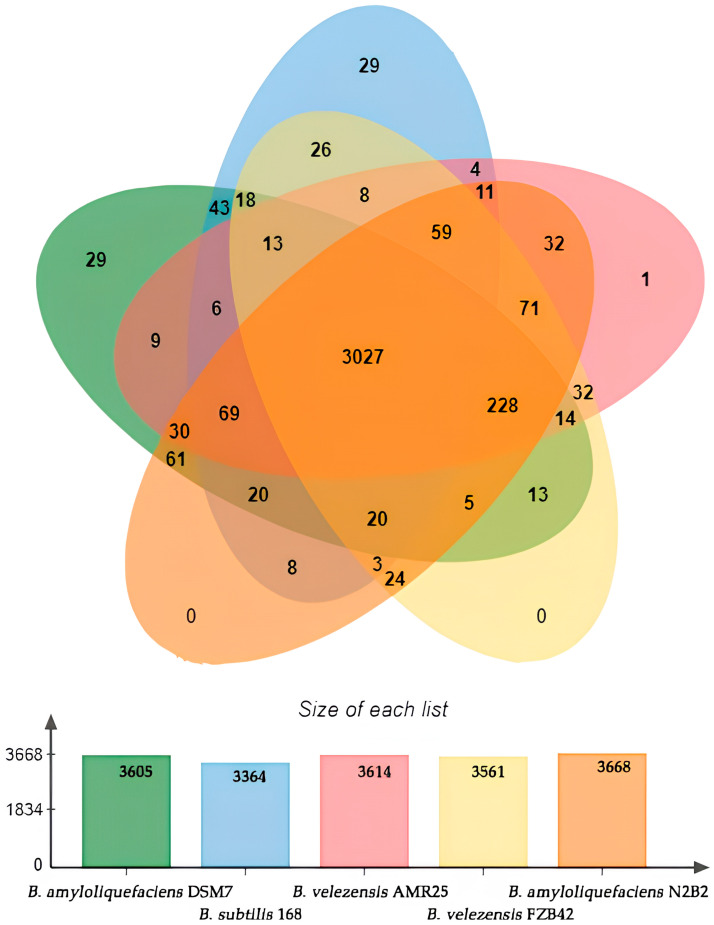
Venn diagram of *Bacillus velezensis* AMR25, *Bacillus velezensis* FZB42, *Bacillus amyloliquefaciens* DSM7, *Bacillus amyloliquefaciens* N2B2, and *Bacillus subtilis* 168. The numbers of gene clusters between the genome subsets.

**Figure 6 microorganisms-12-01533-f006:**
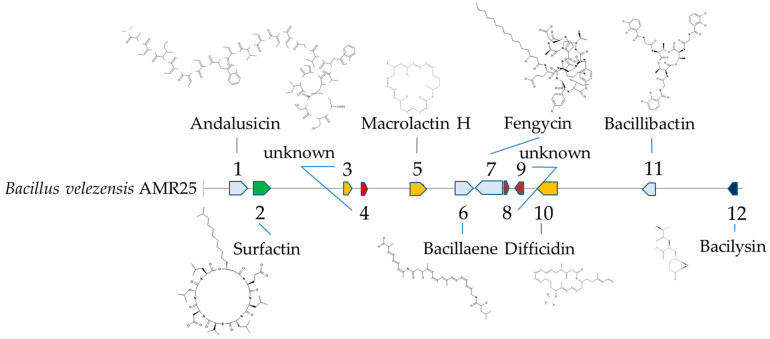
Gene cluster identified in *Bacillus velezensis* AMR25 genome with orientation using antiSMASH software version 7.0.0. 1—andalusicin; 2—surfactin; 3, 4—unknown; 5—macrolactin; 6—bacillaene 7—fengycin; 8, 9—unknown; 10—difficidin; 11—bacillibactin; 12—bacilysin. Additional information on the structure of genetic clusters is provided in Appendix A.

**Figure 7 microorganisms-12-01533-f007:**
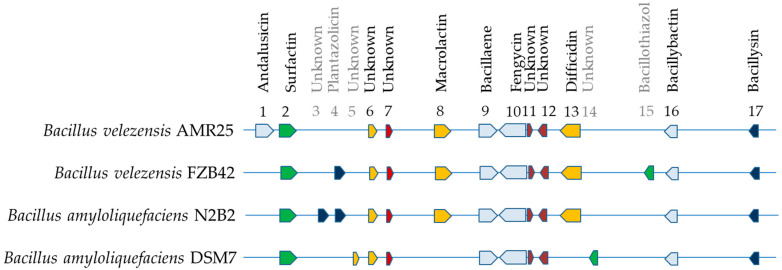
Gene clusters identified in the genome of *Bacillus velezensis* AMR25 and related members of the genus *Bacillus* with orientation using antiSMASH software version 7.0.0. 1—andalusicin; 2—surfactin; 3—unknown; 4—plantazolicin; 5, 6, 7—unknown; 8—macrolactin; 9—bacillaene 10—fengycin; 11, 12—unknown; 13—difficidin; 14—unknown; 15—bacillothiozol; 16—bacillibactin; 17—bacilysin.

**Figure 8 microorganisms-12-01533-f008:**
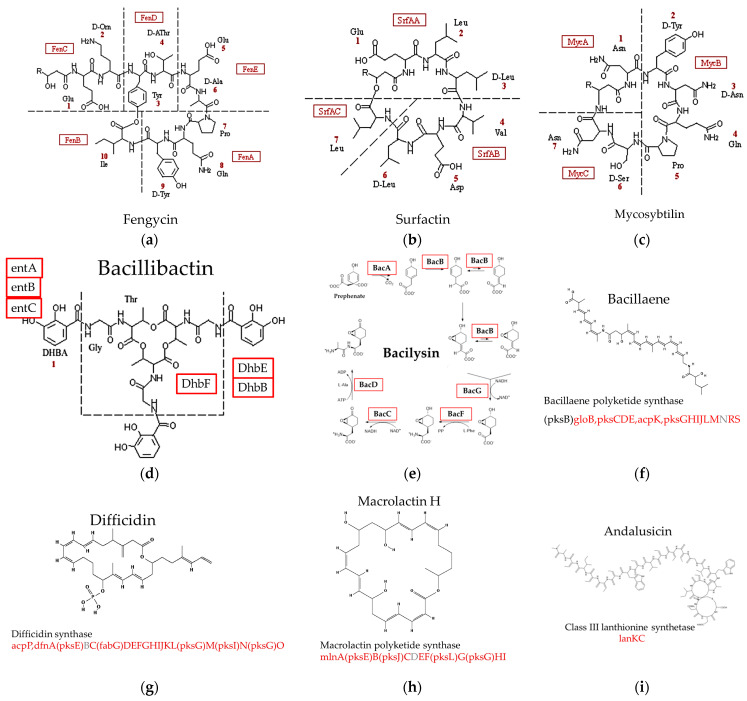
Structural representation and synthetic gene of (**a**) fengycin, (**b**) surfactin and (**c**) mycosubtilin, (**d**) bacilibactin, (**e**) bacilisine, (**f**) bacilaene, (**g**) difficidine, (**h**) macrolactin, (**i**) andalusicine. Proteins whose genes are found in the *Bacillus velezensis* AMR25 genome are highlighted in red while those that could not be identified are indicated in gray.

**Table 1 microorganisms-12-01533-t001:** Area of inhibition (mm) of *Bacillus velezensis* strain AMR25 against some plant bacterial pathogens. The experiments were performed at least three times and the results are represented as mean ± standard error.

Pathogens	*E* *rwinia billingiae*	*P* *antoae agglomerans*	*X* *antomonas campestris*
Area of inhibition (mm)	4.33 ± 0.33	3.66 ± 0.33	1.66 ± 0.33

**Table 3 microorganisms-12-01533-t003:** Genomic features of the *Bacillus velezensis* AMR25 and related members of the *Bacillus* genus.

	AMR25	N2B2	FZB42	DMS7	168
Genome size (bp)	3,909,646	3,953,520	3,918,589	3,980,199	4,214,630
G+C content (mol%)	46.6%	46.6%	46.4%	46.1%	43.5%
Number of protein-coding sequences	3716	3886	3693	3921	4106
Average CDS size (bp)	923	925	933	888	895
Percentage of coding regions	88.50%	88%	88%	87%	87%
Number of tRNA	86	65	89	94	86
Number of rRNA operons	9	1	10	10	10
IS elements	2	2	9	n.r.	0
Phage-associated genes	47	108	44	185	268

AMR25—*Bacillus velezensis* strain AMR25 (CP140115.1); N2B2—*B. amyloliquefaciens* N2B2 (GCA_030284565.1); FZB42—*B. velezensis* FZB42 (GCA_000015785.2); DSM7—*B. velezensis* DSM7 (GCA_000196735.1); 168—*B. subtilis* subsp. *subtilis* 168 (GCA_000009045.1). n.r.—not registered.

**Table 4 microorganisms-12-01533-t004:** Gene clusters involved in synthesis of biocontrol metabolites in *Bacillus velezensis* AMR25 genome, using the antiSMASH 7.0.0 software.

Region (Figure 6)	Metabolite	Most Similar Known Cluster	Size, Kb	Bioactive Spectrum	Similarity
1	Andalusicin A/B	RiPP: Lanthipeptide	83.1	Inhibits the growth of Gram-positive bacteria [44]	100%
2	Surfactin	NRP: Lipopeptide	64.8	*Cochliobolus carbonum*; *Pseudomonas syringae* pv. tomato *Ralstonia solanacearum*; *Fusarium verticillioides* [45,46]	82%
5	Macrolactin H	Polyketide	87.8	*Staphylococcus aureus*; *Burkholderia cepacia* [47]	100%
6	Bacillaene	Polyketide + NRP	81.7	*Erwinia amylovora* [48]	85%
7	Fengycin	NRP	136.8	*R. solanacearum*; *Xanthomonas euvesicatoria*; *F. oxysporum*; *F. graminearum*; *Botrytis cinerea* [49,50,51]	100%
10	Difficidin	Polyketide	93.8	*X. oryzae* pv. oryzae; *E. amylovora*; *R. solanacearum* [46]	100%
11	Bacillibactin	NRP	51.8	*S. aureus*; *Enterococcus faecalis*; *Klebsiella pneumonia* [52]	100%
12	Bacilysin	Other	41.4	*S. aureus*; *X. oryzae* pv. oryzae; *E. amylovora* [46]	100%

**Table 5 microorganisms-12-01533-t005:** Antibiotic resistance of *B*. *velezensis* AMR25: R—resistant, I—intermediate, S—susceptible.

Antibiotic	Disk Concentration (µg/mL)	Area Diam (mm)	Category
**Penicillins**			
Amoxicillin	10	0	R
Oxacillin	1	0	R
Penicillin	10	7	R
**Cephems**			
Cefoperazone	75	12	R
Ceftazidime	30	0	R
**Macrolides**			
Azithromycin	10	13	R
Erythromycin	15	14	R
Oleandomycin	15	1	R
**Fluoroquinolones**			
Levofloxacin	5	18	S
**Aminoglycoside**			
Gentamicin	10	9	R
**Glycopeptides**			
Vancomycin	30	13	I
**Tetracyclines**			
Tetracycline	30	12	R

**Table 6 microorganisms-12-01533-t006:** Antibiotic resistance genes of *Bacillus velezensis* AMR25.

Accession	Start	Stop	Orientation	Gene Name	Resistance Mechanism	Drug Class
ARO: 3002972	509,896	511,065	+	*vanT*	antibiotic target alteration	glycopeptide antibiotic
ARO: 3002814	565,150	566,199	+	*clbA*	antibiotic target alteration	lincosamide antibiotic; streptogramin antibiotic; oxazolidinone antibiotic; phenicol antibiotic; pleuromutilin antibiotic
ARO: 3002877	1,195,572	1,196,486	+	*BcI*	antibiotic inactivation	cephalosporin; penem
ARO: 3007014	1,811,775	1,812,128	−	*qacJ*	antibiotic efflux	disinfecting agents and antiseptics
ARO: 3002956	2,087,420	2,088,238	−	*vanY*	antibiotic target alteration	glycopeptide antibiotic
ARO:3002818	2,124,728	2,125,162	−	*msrB*	antibiotic target protection	streptogramin antibiotic, macrolide antibiotic
ARO:3000251	2,125,164	2,125,697	−	*msrA*	antibiotic target protection	streptogramin antibiotic, macrolide antibiotic
ARO:3004958	2,175,434	2,178,217	+	*ponA*	antibiotic target alteration	penicillin antibiotic
ARO: 3003196	2,554,632	2,556,008	−	*tet(45)*	antibiotic efflux	tetracycline antibiotic
ARO: 3007015	3,883,659	3,884,042	−	*qacG*	antibiotic efflux	disinfecting agents and antiseptics

**Table 2 microorganisms-12-01533-t002:** Inhibition index of *Bacillus velezensis* strain AMR25 against some plant fungal pathogens. The experiments were performed at least three times and the results are represented as mean ± standard error.

Pathogens	*A* *lternaria tenuissima*	*B* *otrytis cinerea*	*F* *usarium avenacea*
Inhibition Index (%)	65.71 ± 1.48%	71.72 ± 1.48%	62.86 ± 1.48%

## Data Availability

The complete genome sequence for *B. velezensis* AMR25 can be found in the following NCBI database “BioProject PRJNA1049968” with accession number CP140115.1, the URL of the genome sequence is: https://www.ncbi.nlm.nih.gov/nuccore/CP140115.1/, (accessed on 22 July 2024).

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
