# Peer review of "Whole Genome Sequencing of Bacillus velezensis AMR25, an Effective Antagonist Strain against Plant Pathogens"

_microorganisms, 2024, doi:10.3390/microorganisms12081533_

Round 1
Reviewer 1 Report
Comments and Suggestions for Authors
The manuscript describes de genome sequencing of an endophytic bacterial strain (Bacillus velezensis strain AMR25) isolated from the wild grapevine Vitis amurensis. According to the inhibitions test B.velezensis AMR25 can inhibit different bacterial and fungal plant pathogens. The genome analysis revealed the presence of genes implicated in the production of different antimicrobial and antifungal molecules, as well as genes implicated in different metabolic pathways related to the production of plant growth-promoting molecules such as indole acetic acid and phytohormones, but also different genes related to antibiotic resistance.
Overall, the manuscript is well-redacted and could be of interest to Microorganisms Journal readers, but some important aspects must be improved.
Authors must address the following commentaries:
Lines 28 and 32, bold font is used in references
Lines 40-43, review redaction for clearer understanding of the information
Lines 51-64, review redaction for clearer understanding of the information, connector are needed for a better reading and comprehension of the information.
Line 58, check format in “phytopathogens, [14–16].”
Line 61-64, complement with information about the usages of the Bacillus-based products
Line 78, use italic for scientific name “B. amyloliquefaciens”
Line 93, use superscript in “10^8”
Line 94, check format in “ap-plied”
Line 100-102, explain in more detail the methodology, in specific how growth inhibition was determined.
Line 115, check format in “26 °C”
Line 117, eliminate comma
Lines 136-137, check format in “The overnight bacterial suspension was centrifuged at +4 °C 6 000 g for 6 min”
Line 185, why use “zone inhibition” instead inhibition diameter, or the inhibition area was determined?
Figure 1, Explain better what is in each panel, photographs in panels a, b, and c are growth references?
Line 194-196, use points instead commas in the inhibition percentages, same in Table 2
In figure 2, give visual reference od how the diameters R1 and R2 were determined, in the photographs fungal growth seems quite similar in both treatments
In figure 3, explain which means percentages, are identities among gene sequences of the antimicrobial molecules?
Line 233, explain what percentages means
Line 242, add bacterial species name, not just strain code
Lines 249-353, add commas in the gene numbers
Line 252, eliminate extra space in “(Figure 5). The”
Line 255, add comma in the gene number
Line 256, use italic for scientific name “B. amyloliquefaciens”
Line 285 and 288, add bacterial species name, not just strain code
Line 296, explain why it was selected to analyze the genes of these compounds and not those of others, mycosubtiilin is not represented in figure 7
Lins 298 and 298, use a synonym in the second “structure”
Respect antibiotic resistance determination, explain why determine it, and which are the implications of the results get from this experiment
Line 377, choose other terms for “drugs”
Line 389, use “%” instead “percentage!
Line 394 and 395, check “mycosubtiilin” is repeated
Lines 408-410, add bacterial species name, not just strain code
Line 410, use point instead comma in percentage
In conclusions include aspects related to the bacterial and fungal plant pathogens inhibition and antibiotic resistance that also were evaluated in the study
Author Response
Dear Reviewer! We would like to express our gratitude for your thorough review of our manuscript. Your comments and suggestions have significantly improved the quality and relevance of our work. We appreciate your attention to detail and constructive criticism, which has helped us to enhance the overall impact of our research. Thank you for taking the time to provide us with valuable feedback. Best regards, Dr. Ogneva. On behalf of the authoring team.
- Lines 28 and 32, bold font is used in references
Response: Thanks for your comment, we fixed it.
- Lines 40-43, review redaction for clearer understanding of the information
Response: Thanks for your comment, we fixed it.
- Lines 51-64, review redaction for clearer understanding of the information, connector are needed for a better reading and comprehension of the information.
Response: Thanks for your comment, we fixed it.
- Line 58, check format in “phytopathogens, [14–16].”
Response: Thanks for your comment, we fixed it.
- Line 61-64, complement with information about the usages of the Bacillus-based products
Response: Thanks for your comment, we fixed it.
- Line 78, use italic for scientific name “B. amyloliquefaciens”
Response: Thanks for your comment, we fixed it.
- Line 93, use superscript in “10^8”
Response: Thanks for your comment, we fixed it.
- Line 94, check format in “ap-plied”
Response: Thanks for your comment, we fixed it.
- Line 100-102, explain in more detail the methodology, in specific how growth inhibition was determined.
Response: Thanks for your comment, we fixed it.
- Line 115, check format in “26 °C”
Response: Thanks for your comment, we fixed it.
- Line 117, eliminate comma
Response: Thanks for your comment, we fixed it.
- Lines 136-137, check format in “The overnight bacterial suspension was centrifuged at +4 °C 6 000 g for 6 min”
Response: Thanks for your comment, we fixed it.
- Line 185, why use “zone inhibition” instead inhibition diameter, or the inhibition area was determined?
Response: Thanks for your comment, we fixed it.
- Figure 1, Explain better what is in each panel, photographs in panels a, b, and c are growth references?
Response: Thanks for your comment, we have added this information to the text.
- Line 194-196, use points instead commas in the inhibition percentages, same in Table 2
Response: Thanks for your comment, we fixed it.
- In figure 2, give visual reference od how the diameters R1 and R2 were determined, in the photographs fungal growth seems quite similar in both treatments
Response: Thanks for your comment; an example of radius calculation is provided in the supplementary files.
- In figure 3, explain which means percentages, are identities among gene sequences of the antimicrobial molecules?
Response: Thanks for your comment, we have added this information to the text.
- Line 233, explain what percentages means
Response: Thanks for your comment, we fixed it.
- Line 242, add bacterial species name, not just strain code
Response: Thanks for your comment, we fixed it.
- Lines 249-353, add commas in the gene numbers
Response: Thanks for your comment, we fixed it.
- Line 252, eliminate extra space in “(Figure 5). The”
Response: Thanks for your comment, we fixed it.
- Line 255, add comma in the gene number
Response: Thanks for your comment, we fixed it.
- Line 256, use italic for scientific name “B. amyloliquefaciens”
Response: Thanks for your comment, we fixed it.
- Line 285 and 288, add bacterial species name, not just strain code
Response: Thanks for your comment, we fixed it.
- Line 296, explain why it was selected to analyze the genes of these compounds and not those of others, mycosubtiilin is not represented in figure 7
Response: We discovered a cluster unique to B. velezensis AMR 25, consisting of 3 genes, which was not detected in the antiSMASH analysis, which present on Figure 7. The detected genes correspond to mycosubtilin synthetase subunits A, B and C, respectively.
- Lins 298 and 298, use a synonym in the second “structure”
Response: Thanks for your comment, we fixed it.
- Respect antibiotic resistance determination, explain why determine it, and which are the implications of the results get from this experiment
Response: The determination of antibiotic resistance is a crucial aspect of the study of a novel strain of endophytic organism, as it allows for a better understanding of its potential pathogenicity. Under certain conditions, this strain may exhibit pathogenic properties, necessitating knowledge of which antibiotics it naturally possesses resistance to. Furthermore, when employing bacterial strains for biocontrol purposes, it is essential to be aware of their antibiotic resistance profiles in order to assess their ability to thrive in the presence of pathogenic microorganisms such as fungi that secrete large quantities of antibiotics into their environment.
- Line 377, choose other terms for “drugs”
Response: Thanks for your comment, we fixed it.
- Line 389, use “%” instead “percentage!
Response: Thanks for your comment, we fixed it.
- Line 394 and 395, check “mycosubtiilin” is repeated
Response: Thanks for your comment, we fixed it.
- Lines 408-410, add bacterial species name, not just strain code
Response: Thanks for your comment, we fixed it.
- Line 410, use point instead comma in percentage
Response: Thanks for your comment, we fixed it.
Reviewer 2 Report
Comments and Suggestions for Authors
The presented research may be important in expanding theoretical and practical knowledge regarding biocontrol in plant cultivation. A special form of biocontrol is the endophytic microbiome. The authors of the study attempted to investigate the endophytic bacterium Bacillus velezensis strain AMR25 as biological control agents against plant pathogens. They suggest that this bacterium could be used as biocontrol of plant diseases caused by fungi. The authors conducted the research correctly and presented and discussed the results extensively.
Line 93: night cultures? Please explain this
Author Response
Dear Reviewer! We would like to express our gratitude for your thorough review of our manuscript. Your comments and suggestions have significantly improved the quality and relevance of our work. We appreciate your attention to detail and constructive criticism, which has helped us to enhance the overall impact of our research. Thank you for taking the time to provide us with valuable feedback. Best regards, Dr. Ogneva. On behalf of the authoring team.
- Line 93: night cultures? Please explain this
Response: Overnight cultures will be right. Thank you, for your comment.
Reviewer 3 Report
Comments and Suggestions for Authors
The manuscript described the identification of a Bacillus velezensis AMR25 isolated from Vitis amurensis Rupr. that exhibited antimicrobial activity against plant bacterial pathogens, including Erwinia billingiae, Pantoae agglomerans, and Xantomonas campestris, and fungal pathogens, including Alternaria tenuissima, Botrytis cinerea, and Fusarium avenacea. The B. velezensis AMR25 genome was sequenced, and gene clusters involved in the synthesis of antimicrobial substances and genes related to plant-bacterial interaction were analyzed. The potential use of B. velezensis AMR25 for the development of biocontrol products was assumed. It is recommended for the journal Microorganisms after further modifications.
Major:
1. The bacterial pathogens isolated from Vitis amurensis Rupr. were described be added to the Federal Scientific Center for Biodiversity in Section 2.2. However, the institute could not be found on the website. The correct name of the institute and its website link need to be shown. In addition, the isolate code and the center code for each bacterial pathogen also have to be shown.
2. Were the fungal pathogens isolated from Vitis amurensis Rupr. also deposited in the same institute? The isolate code and the center code for each fungal pathogen also have to be shown.
3. The data of the B. velezensis AMR25 growth condition and the zone of inhibition shown in Figure 1 was inconsistent. It seemed B. velezensis AMR25 grew for a longer time in Figure 1F than 1E, and then 1D.
4. The phylogenetic tree, Figure 4, built according to the gyrA partial gene from various Bacillus sp. showed the studied B. velezensis AMR25 was questioned. Four Bacillus sp. were 100% the same as B. velezensis AMR25, but the bootstrap value was 97. The reasons could be explained in the text.
5. The accession numbers of the Bacillus sp. genome sequences should be described in Section 2.7, and the legend of Figure 3 and the footnote of Table 3.
6. The numbers of CDS shown in Figure 5 didn’t match to those shown in Table 3.
7. The start, the stop, and the orientation of the biocontrol gene clusters can be indicated in Figure 6 and Table 4.
8. How to get the data of similarity shown in Table 4 should be described in the text and the footnote of Table 4. In addition, some of the bioactive spectrum didn’t provide references.
9. The start, the stop, and the orientation of the biocontrol gene clusters of three Bacillus sp. can be indicated in Figure 7. The names of the secondary metabolites 2, 4, 5, and 9, according to B. amyloliquefaciens N2B2, were not shown. In addition, B. amyloliquefaciens DSM7 could be analyzed for consistency.
10. The presence of all the genes necessary for the biosynthesis of fengycin, surfactin, and mycosubtilin in the cluster was analyzed and shown in Figure 8. The other five could also be analyzed.
11. Section 3.6. According to Tables 5 and 6, the genes responsible for several antibiotic-resistant strains were not found. This could be further discussed in the text.
12. Section 4. The protein numbers involved in chemotaxis and flagellar formation, and gene numbers involved in adhesion and biofilm formation differed from those shown in Table S1.
13. The start, the stop, the gene length, and the orientation of the genes shown in Table S1 could be indicated.
Minor:
1. The circular map shown in Figure 3 could be enlarged to show the details. An additional table could indicate the start, the stop, and the orientation of the biocontrol gene clusters, as shown for antibiotic-resistant genes in Table 6.
2. The accession numbers of the gyrA partial gene from various Bacillus sp. used to build the phylogenetic relationship must be shown in Section 2.5 and the legend of Figure 4. The multiple sequence alignment could be shown in a supplementary figure.
3. A typo, FZ42, was found in Table 3. The bacterial scientific name should be in italics. In addition, the definition of n. r. should be indicated.
4. Section 3.7.2. The Table S1 was not mentioned.
Author Response
Dear Reviewer! We would like to express our gratitude for your thorough review of our manuscript. Your comments and suggestions have significantly improved the quality and relevance of our work. We appreciate your attention to detail and constructive criticism, which has helped us to enhance the overall impact of our research. Thank you for taking the time to provide us with valuable feedback. Best regards, Dr. Ogneva. On behalf of the authoring team.
Major:
- The bacterial pathogens isolated from Vitis amurensis Rupr. were described be added to the Federal Scientific Center for Biodiversity in Section 2.2. However, the institute could not be found on the website. The correct name of the institute and its website link need to be shown. In addition, the isolate code and the center code for each bacterial pathogen also have to be shown.
Response: Thank you for your comment. However, our scientific institute does not have a strain bank or maintain a collection for a long period of time. Therefore, we are unable to provide you with an isolate code or a center code for each bacterial pathogen. The pathogens were isolated from the tissues of infected plants, then identified using molecular biological techniques and were immediately used in experiments. Federal Scientific Center of the East Asia Terrestrial Biodiversity, FEB RAS, 690022 Vladivostok, Russia. Website: https://www.biosoil.ru/
- Were the fungal pathogens isolated from Vitis amurensis Rupr. also deposited in the same institute? The isolate code and the center code for each fungal pathogen also have to be shown.
Response: Thank you for your comment. The pathogens were isolated from the tissues of infected plants, then identified using molecular biological techniques and were immediately used in experiments.
- The data of the B. velezensisAMR25 growth condition and the zone of inhibition shown in Figure 1 was inconsistent. It seemed B. velezensis AMR25 grew for a longer time in Figure 1F than 1E, and then 1D.
Response: Thanks for your comment. In this experiment, the same time was used for all six Petri dishes shown in Figure 1. Petri dish A - Erwinia billingiae; Petri dish B - Pantoae agglomerans; Petri dish C - Xantomonas campestris, are represented, respectively on dishes D, E, F - agarose blocks containing Bacillus velezensis AMR25 were added to above pathogens. The observed difference in the size of the bacterial mass between Petri dishes D, E and F is explained by the fact that, with such a test, there is an interaction between the growth parameters of the pathogen and strain under study.
- The phylogenetic tree, Figure 4, built according to the gyrApartial gene from various Bacillus sp. showed the studied B. velezensis AMR25 was questioned. Four Bacillus sp. were 100% the same as B. velezensis AMR25, but the bootstrap value was 97. The reasons could be explained in the text.
Response: Thanks for the comment. I didn't quite understand the question correctly. The gyrA gene is in the public domain. The link to it is provided in the manuscript at coordinates CP140115.1:6595-9054. Using the BLAST algorithm, the only organism with a 100% similarity was Bacillus velezensis AMR25. A phylogenetic tree was constructed using the MEGAX program and gyrA sequences from Bacillus strains closest to the 16S rRNA sequence.
- The accession numbers of the Bacillussp. genome sequences should be described in Section 2.7, and the legend of Figure 3 and the footnote of Table 3.
Response: Thank you for your comment. I have added the data where you requested it.
- The numbers of CDS shown in Figure 5 didn’t match to those shown in Table 3.
Response: Thank you for your comment. Indeed, there was a misunderstanding. Table 3 shows the number of protein-coding regions that were calculated during genome annotation using the Genome Automatic Annotation Pipeline (PGAAP). In Figure 5 Venn diagram, the OrthoVenn was used to find the intersection of gene clusters between genomes. The caption of Figure 5 has been updated.
- The start, the stop, and the orientation of the biocontrol gene clusters can be indicated in Figure 6 and Table 4.
Response: Thank you for your comment. This data has been formatted into a table S1. In the supplementary files, you will find all the relevant information.
- How to get the data of similarity shown in Table 4 should be described in the text and the footnote of Table 4. In addition, some of the bioactive spectrum didn’t provide references.
Response: The similarity data was obtained using the antiSMASH algorithm. References have been added, thanks for your care.
- The start, the stop, and the orientation of the biocontrol gene clusters of three Bacillussp. can be indicated in Figure 7. The names of the secondary metabolites 2, 4, 5, and 9, according to B. amyloliquefaciens N2B2, were not shown. In addition, B. amyloliquefaciens DSM7 could be analyzed for consistency.
Response: Thanks for your comment; the figure 7 has been changed to a more detailed one using the DSM7 strain
- The presence of all the genes necessary for the biosynthesis of fengycin, surfactin, and mycosubtilin in the cluster was analyzed and shown in Figure 8. The other five could also be analyzed.
Response: Thank you for your comment. In the supplementary files, you will find all the relevant information about genes necessary for the biosynthesis.
- Section 3.6. According to Tables 5 and 6, the genes responsible for several antibiotic-resistant strains were not found. This could be further discussed in the text.
Response: Yes, this is a valid point, but we attribute this phenomenon to the fact that, when analyzing the genome of Bacillus velezensis AMR25, we only had chromosome sequencing data available, which did not include information about plasmid sequences. These plasmids could potentially be responsible for the resistance to these antibiotics, and in order to provide an accurate Response, it would be necessary to obtain complete plasmid sequence information.
- Section 4. The protein numbers involved in chemotaxis and flagellar formation, and gene numbers involved in adhesion and biofilm formation differed from those shown in Table S1.
Response: We have resolved the issue.
- The start, the stop, the gene length, and the orientation of the genes shown in Table S1 could be indicated.
Response: The data has been added to the table S1.
Minor:
- The circular map shown in Figure 3 could be enlarged to show the details. An additional table could indicate the start, the stop, and the orientation of the biocontrol gene clusters, as shown for antibiotic-resistant genes in Table 6.
Response: The data has been added to the table S1.
- The accession numbers of the gyrApartial gene from various Bacillus sp. used to build the phylogenetic relationship must be shown in Section 2.5 and the legend of Figure 4. The multiple sequence alignment could be shown in a supplementary figure.
Response: The data has been added.
- A typo, FZ42, was found in Table 3. The bacterial scientific name should be in italics. In addition, the definition of n. r. should be indicated.
Response: We have resolved the issue.
- Section 3.7.2. The Table S1 was not mentioned.
Round 2
Reviewer 1 Report
Comments and Suggestions for Authors
After carefully reviewing the new version of the manuscript, I consider the authors well addressed the prior commentaries and suggestions of the reviewers, I don´t have more commentaries.
Author Response
Comments 1: After carefully reviewing the new version of the manuscript, I consider the authors well addressed the prior commentaries and suggestions of the reviewers, I don´t have more commentaries.
Answer: Thank you for your valuable contribution to my manuscript.
With regard to your work,
Dr. Ogneva.
Reviewer 3 Report
Comments and Suggestions for Authors
The manuscript described the identification of a Bacillus velezensis AMR25 isolated from Vitis amurensis Rupr. that exhibited antimicrobial activity against three plant bacterial pathogens and three fungal pathogens. The genome was sequenced, and gene clusters involved in the synthesis of antimicrobial substances and genes related to plant-bacterial interaction were analyzed. Though the manuscript was revised, many suggested improvements were not neglected and remained unsolved. It is, therefore, not recommended for the journal Microorganisms. The unaddressed questions and suggestions are shown below.
Major:
1. Section 2.1 did not show the correct institute name and website link. In addition, the author replied that the institute didn’t maintain the bacterial collections, making the current study unproducible and cannot be re-evaluated.
2. The fungal pathogens isolated from Vitis amurensis Rupr. were not maintained available. It has the same doubts.
3. The data of the B. velezensis AMR25 growth condition and the zone of inhibition shown in Figure 1 were inconsistent. It seemed B. velezensis AMR25 grew for a longer time in Figure 1F than 1E, and then 1D. The authors replied that there is an interaction between the growth parameters of the pathogen and the strain under study. Another method could be used to solve this problem and confirm the elucidation that the authors suggested.
4. The phylogenetic tree, Figure 4, built according to the gyrA partial gene from various Bacillus sp. Were four Bacillus sp. 100% the same as B. velezensis AMR25 because the bootstrap value was 97? The multiple sequence alignment was not provided, either. The reasons could be explained in the text.
5. The numbers of CDS shown at the bottom of Figure 5 didn’t match to those shown in Table 3. Using gene clusters to replace the CDS in the revised version was incorrect.
6. Figure 6 and Table 4 did not indicate the start, stop, and orientation of the biocontrol gene clusters, though a detailed supplementary Table S1 was provided. Brief information shown in the figure would be necessary.
7. The authors described using antiSMASH 7.0 to get the similarity data shown in Table 4. However, the details were not described, including 1) Were full-length gene cluster sequences used? 2) Were sequences from four similar Bacillus sp. used for comparison or one of them? They should be described in the text, the footnote of Table 4, and Section 2.8. In addition, 100% is identical, not similar.
8. Table 4. The bioactive spectrum of region 11, microbial competitors, didn’t provide references. It should be further explained.
9. The start, stop, and orientation of the biocontrol gene clusters of three Bacillus sp. can be indicated in Figure 7. Brief information shown in the figure would be necessary.
10. Several names of the secondary metabolites were not shown in Figure 7. They should be described in the legend even if they were unknown.
11. The presence of all the genes necessary for the biosynthesis of the other five metabolites, in addition to fencing, surfactin, and mycosubtilin, could also be analyzed, as illustrated in Figure 8.
12. Section 3.6. According to Tables 5 and 6, the genes responsible for several antibiotic-resistant strains were not found. The authors explained that they could be on the plasmids. The text could further discuss this, and the plasmid-mediated antibiotic resistance could be analyzed.
13. Section 4. The protein numbers involved in chemotaxis and flagellar formation and gene numbers involved in adhesion and biofilm formation differed from those shown in Table S1. The authors claimed the issue was solved, but it was not.
Minor:
1. The circular map shown in Figure 3 could be enlarged to show the details. The descriptions on the right could be displayed at the bottom.
2. The multiple sequence alignment of the gyrA partial gene from various Bacillus sp. used to build the phylogenetic relationship can be shown in a supplementary figure.
3. A typo, FZ42, was found in Table 3. It was unsolved.
Author Response
Major:
Comment 1. Section 2.1 did not show the correct institute name and website link. In addition, the author replied that the institute didn’t maintain the bacterial collections, making the current study unproducible and cannot be re-evaluated.
Answer: Our institute does not maintain a publicly accessible database of its collection. But used AMR25 strain is kept in the collection of our laboratory. Our collection contains approximately 200 strains of endophytes and 20 strains of plant pathogens. Upon request, the laboratory can provide the strains used in the study. Furthermore, the name of the institute has been amended in accordance with the correction, which can be verified on line 77.
Comment 2. The fungal pathogens isolated from Vitis amurensis Rupr. were not maintained available. It has the same doubts.
Answer: Our institute does not maintain a publicly accessible database of its collection. But used AMR25 strain is kept in the collection of our laboratory. Our collection contains approximately 200 strains of endophytes and 20 strains of plant pathogens. Upon request, the laboratory can provide the strains used in the study. Furthermore, the name of the institute has been amended in accordance with the correction, which can be verified on line 77.
Comment 3. The data of the B. velezensis AMR25 growth condition and the zone of inhibition shown in Figure 1 were inconsistent. It seemed B. velezensis AMR25 grew for a longer time in Figure 1F than 1E, and then 1D. The authors replied that there is an interaction between the growth parameters of the pathogen and the strain under study. Another method could be used to solve this problem and confirm the elucidation that the authors suggested.
Answer: Regarding the growth of bacteria, it is important to understand that different types of bacteria grow on different substrates. In other words, the rate of growth of a particular bacterium may vary depending on the environment in which it is growing. If a bacterium is grown on a specific type of substrate, such as Petri dishes full of pathogenic bacteria, it may produce different substances that can affect the growth rate of other bacteria. This is because each type of bacterium has its own unique set of genes and enzymes that allow it to survive and thrive in its specific environment. Therefore, it is not possible to predict how a bacterium will grow in the presence of other microorganisms, as each bacterium has a unique response to its surroundings.
Comment 4. The phylogenetic tree, Figure 4, built according to the gyrA partial gene from various Bacillus sp. Were four Bacillus sp. 100% the same as B. velezensis AMR25 because the bootstrap value was 97? The multiple sequence alignment was not provided, either. The reasons could be explained in the text.
Answer: We took your feedback into account and Figure S1 showing a multiple alignment of the gyrA gene. The sequences used have also been added to Table S3. Please see lines 251-253.
Comment 5. The numbers of CDS shown at the bottom of Figure 5 didn’t match to those shown in Table 3. Using gene clusters to replace the CDS in the revised version was incorrect.
Answer: A reexamination of the data revealed an error in the number of coding sequences. This error has been corrected in Table 3. The disparity in the number of CDS identified utilising PGAAP and OrthoVenn is attributable to the distinctive characteristics of their respective algorithms. OrthoVenn combines redundant protein-coding sequences. Please refer to Table 3 for details of the changes and line 225.
Comment 6. Figure 6 and Table 4 did not indicate the start, stop, and orientation of the biocontrol gene clusters, though a detailed supplementary Table S1 was provided. Brief information shown in the figure would be necessary.
Answer: Due to the large amount of information, the drawing would become overloaded, so cluster orientations have been added and the coordinates can be found in supplementary files.
Comment 7. The authors described using antiSMASH 7.0 to get the similarity data shown in Table 4. However, the details were not described, including 1) Were full-length gene cluster sequences used? 2) Were sequences from four similar Bacillus sp. used for comparison or one of them? They should be described in the text, the footnote of Table 4, and Section 2.8. In addition, 100% is identical, not similar.
Answer: The entire genome was utilized for analysis in antiSMASH 7.0, and the program was compared with comprehensive sequences of biosynthesis gene classes accessible within its databases - Minimum Information about a Biosynthetic Gene cluster (MiBIG), Protein Families Database (Pfam). Of the most closely related strains, only one, N2B2, was selected for analysis, as it was also an endophyte and exhibited the most intriguing characteristics. AntiSMASH report an identity, and thus, this value was retained. We indicated that the genome was used, and corrections can be found on lines 189 and 310.
Comment 8. Table 4. The bioactive spectrum of region 11, microbial competitors, didn’t provide references. It should be further explained.
Answer: Thanks for the comment; we have added a link to the literary source in Table 4.
Comment 9. The start, stop, and orientation of the biocontrol gene clusters of three Bacillus sp. can be indicated in Figure 7. Brief information shown in the figure would be necessary.
Answer: Due to the large amount of information, the drawing would become overloaded, so cluster orientations have been added and the coordinates can be found in supplementary files.
Comment 10. Several names of the secondary metabolites were not shown in Figure 7. They should be described in the legend even if they were unknown.
Answer: Figures 6 and 7 have been revised to reflect the comment and the cluster orientations have been added. The names of the secondary metabolites have also been added and are shown in Figure 7 and described in the caption, including those that are unknown.
Comment 11. The presence of all the genes necessary for the biosynthesis of the other five metabolites, in addition to fencing, surfactin, and mycosubtilin, could also be analyzed, as illustrated in Figure 8.
Answer: This analysis has been added to the text at line 337-353 of Figure 8 d-k, and to a Table S2 in supplementary files.
Comment 12. Section 3.6. According to Tables 5 and 6, the genes responsible for several antibiotic-resistant strains were not found. The authors explained that they could be on the plasmids. The text could further discuss this, and the plasmid-mediated antibiotic resistance could be analyzed.
Answer: A more comprehensive examination of the prevalence of antibiotic resistance genes has been conducted. The observed changes are presented in lines 368-379. We have shown the presence of resistance genes for almost all antibiotics to which the AMR25 strain was resistant. The reasons for the absence of gentamicin resistance genes in the AMR25 genome have been discussed in the Discussion section, see lines 473-502.
Comment 13. Section 4. The protein numbers involved in chemotaxis and flagellar formation and gene numbers involved in adhesion and biofilm formation differed from those shown in Table S1. The authors claimed the issue was solved, but it was not.
Answer: The note has been corrected and the note can be found on line 504-506.
Minor:
Comment 1. The circular map shown in Figure 3 could be enlarged to show the details. The descriptions on the right could be displayed at the bottom.
Answer: The image has been amended in accordance with the editorial instructions. Please refer to Figure 3 for further details.
Comment 2. The multiple sequence alignment of the gyrA partial gene from various Bacillus sp. used to build the phylogenetic relationship can be shown in a supplementary figure.
Answer: Following your feedback, Figure S1 shows a multiple alignment of the gyrA gene. The sequences used have also been added to Table S3.
Comment 3. A typo, FZ42, was found in Table 3. It was unsolved.
Answer: Thank you for your valuable comment. The typo has been corrected, and the corrected text can be found in the corresponding tables.